# Primary tumor-induced immunity eradicates disseminated tumor cells in syngeneic mouse model

Raziye Piranlioglu [1], EunMi Lee[1], Maria Ouzounova[2], Roni J. Bollag [1], Alicia H. Vinyard[1], Ali S. Arbab[1], Daniela Marasco[3], Mustafa Guzel[4], John K. Cowell[1], Muthushamy Thangaraju[1], Ahmed Chadli[1], Khaled A. Hassan[5], Max S. Wicha[5], Esteban Celis[1] & Hasan Korkaya [1]

Although clinically apparent metastasis is associated with late stages of cancer development, micro-metastatic dissemination may be an early event. However, the fate of these early disseminated tumor cells (DTC) remains elusive. We show that despite their capacity to disseminate into secondary organs, 4T1 tumor models develop overt metastasis while EMT6-tumor bearing mice clear DTCs shed from primary tumors as well as those introduced by intravenous (IV) injection. Following the surgical resection of primary EMT6 tumors, mice do not develop detectable metastasis and reject IV-injected tumor cells. In contrast, these cells readily grow and metastasize in immuno-deficient athymic or Rag2$^{-/-}$ mice, an effect mimicked by CD8$^+$ T-cell depletion in immunocompetent mice. Furthermore, recombinant G-CSF or adoptive transfer of granulocytic-MDSCs isolated from 4T1 tumor-bearing mice, induce metastasis by suppressing CD8$^+$ T-cells in EMT6-primed mice. Our studies support the concept of immune surveillance providing molecular insights into the immune mechanisms during tumor progression.

[1] Georgia Cancer Center, Department of Biochemistry and Molecular Biology, Augusta University, 1410 Laney Walker Blvd. CN2136, Augusta, GA 30912, USA. [2] Cancer Research Center of Lyon, 28 Rue Laennec, 69008 Lyon, France. [3] Department of Pharmacy, University of Naples "Federico II", 80134 Naples, Italy. [4] Regenerative and Restorative Research Center (REMER), Medipol University, Kavacık Mah. Ekinciler Cad. No.19 Kavacık Kavşağı - Beykoz, 34810 İstanbul Istanbul, Turkey. [5] Comprehensive Cancer Center, University of Michigan, 1500 E. Medical Center Dr. Ann Arbor, MI 48109, USA. Correspondence and requests for materials should be addressed to H.K. (email: hkorkaya@augusta.edu)

It is widely thought that tumor cells disseminate from a primary site into the circulation during the early stages of tumor development. However, very few tumor cells reach secondary organs and even fewer successfully colonize; thus, the development of clinically significant metastases occurs at late stages of disease[1–3]. Thus, metastatic colonization is an extremely inefficient process partly because the majority of the disseminated tumor cells (DTCs) are eliminated by diverse mechanisms, either in circulation or at secondary sites[1,3]. Although the precise fate of DTCs remains largely unknown, three possible mechanisms have been proposed: (i) DTCs evade immune responses and establish secondary tumors[4,5], (ii) remain dormant as a solitary or micrometastasis via immunoediting[6–8], or (iii) are eliminated by the innate and adaptive immune surveillance[3,9]. Recent reports demonstrated that early DTCs give rise to metastatic colonization in HER2-driven mouse mammary tumor models supporting the tumor dormancy model[10,11]. However, data on the fate of DTCs following the removal of the primary tumor have been conflicting. Retrospective clinical studies suggest that complete resection of primary tumor significantly improves survival in breast cancer patients[12–14]. In contrast, studies performed in mouse models have demonstrated that surgical removal of primary tumors accelerates growth of DTCs at metastatic sites[15–18]. A systemic inflammatory response to surgery was suggested to be one possible mechanism to promoting outgrowths[18]. In other mouse models, however, it has been shown that resection of primary tumor may improve survival by reducing the tumor burden or via reversal of immunosuppression[19,20].

Consistent with Paget's "seed and soil" hypothesis, the term "pre-metastatic niche" has recently been introduced to describe the tumor-induced permissive microenvironment, "soil" in distant organs[21–23]. Accordingly, some tumor cells, "seed" successfully prime the target organs to create a metastatic site, "soil" prior to metastatic spread[23]. In line with the concept, we recently demonstrated that infiltration of a granulocytic subset of myeloid-derived suppressor cells (gMDSC) in lungs creates such pre-metastatic niches in 4T1 tumor-bearing mice[24]. In this model, gMDSCs not only suppress antitumor immunity, but they also promote the epithelial phenotype of cancer stem cells (CSC), which were indeed shown to be proliferative[25]. Using the syngeneic mouse models, we demonstrated here that orthotopically implanted EMT6 tumors fail to generate spontaneous metastasis despite the existence of disseminated solitary or micrometastatic tumor cells in distant organs. However, DTCs or micrometastases progress to full-blown metastasis in 4T1 tumor-bearing mice, resulting in shorter survival[24]. In line with the immune-surveillance concept, we show that EMT6 tumors in syngeneic BALB/c mice induce antitumor immunity, resulting in clearance of DTCs in distant organs. This clearance was mediated by cytotoxic T lymphocytes (CTL), but not by natural killer (NK) cells, as previously reported[26,27]. Furthermore, mice are cured and free of DTCs when primary tumors are completely resected, while mice with residual tumors following surgery show enhanced growth of recurrent primary tumors and concomitant growth of DTCs at the metastatic site.

## Results

**Primary EMT6 tumor-induced antitumor immunity clears DTCs.** We previously characterized murine mammary tumors in their respective syngeneic models and demonstrated that 4T1 tumors develop spontaneous metastasis, while EMT6 tumors fail to do so[24]. We determined that despite their inability to establish metastases, EMT6 tumors indeed disseminate from the primary site into regional lymph nodes and lungs as early as 1 week post implantation, as efficiently as the 4T1 tumors (Fig. 1a–d and

Supplementary Fig. 1). In order to examine the fate of these DTCs in EMT6 tumor-bearing mice, we injected the luciferase-expressing EMT6-Luc cells (100K, via the tail vein) into naive and EMT6 tumor-bearing mice and monitored their growth in the lungs of live animals. Expectedly, EMT6-Luc cells were able to generate pulmonary metastasis in naive mice due to the sheer number of cells; however, they were eliminated in EMT6 tumor-bearing mice (Fig. 1e, f). This may be consistent with the majority of EMT6 tumor-bearing mice surviving longer with large tumors compared with 4T1 tumor-bearing mice that succumbed to death due to extensive metastasis within 6–8 weeks. To determine the specificity of EMT6-induced immune response, we injected EMT6-Luc or 4T1-Luc cells into either EMT6 tumor-bearing or 4T1 tumor-bearing mice. While EMT6 tumor-bearing mice failed to eliminate the 4T1-Luc cells, 4T1 tumor-bearing mice promoted the pulmonary growth of both EMT6-Luc and 4T1-Luc cells when injected via the tail vein and all these animals failed to survive for more than 30 days (Supplementary Fig. 2a–d). We then resected primary EMT6 tumors after 4 weeks post implantation to determine whether the immune clearance of DTCs would occur in the absence of primary EMT6 tumors. Hereafter, these mice are called "EMT6 tumor-primed" mice. EMT6-Luc cells (100K) were injected in EMT6 tumor-primed mice 2 days post resection, and cells trapped in the lungs at 1–2 h post injection were confirmed (Fig. 1g, h). However, EMT6-Luc cells were completely cleared within 10 days, and the animals remained tumor-free for up to 180 days (Fig. 1g, h). These findings prompted us to examine the nature and duration of the antitumor immunity in EMT6 tumor-primed animals. To determine whether EMT6 tumor-primed mice exhibit an immunological memory, EMT6-Luc tumor cells (100K) were injected 3 weeks post resection of primary EMT6 tumors and verified by optical imaging in the lungs after 1–2 h (Fig. 1i, j). All animals repeatedly cleared the EMT6-Luc tumor cells injected via the tail vein after 3, 5, and 9 weeks post resection of primary tumors (Fig. 1i, j). Collectively, these results suggest that EMT6-primed animals may have developed immunological memory against EMT6 tumor cells, as previously reported.

**The antitumor immunity against DTCs is CD8 T-cell dependent.** The observation that primary EMT6 tumor elicits an antitumor immunity that eliminates DTCs in syngeneic mice, suggests that this could be CTL mediated. To examine this, we utilized both athymic nude and Rag2$^{-/-}$ mice, both of them lacking mature T lymphocytes. As expected, orthotopically implanted primary EMT6 tumors in athymic nude or Rag2$^{-/-}$ mice failed to elicit an antitumor immune response, and as a result, these mice developed pulmonary metastasis when EMT6-Luc cells were injected via the tail vein (Fig. 2a–d). Consistent with our previous report, we found significantly higher granulocytic MDSC infiltration in the lungs of primary EMT6 tumor-bearing mice compared with mice without primary tumors (Fig. 2e). Primary tumor-bearing mice also exhibited significantly enlarged spleens (Fig. 2f, g), which we previously observed in metastatic 4T1 tumor-bearing mice[24]. Although our data provide evidence of immune-mediated clearance of DTCs in distant organs, we reasoned that the depletion of CD8$^+$ T cells in a syngeneic model would provide more direct evidence.

To evaluate the effects of CD8$^+$ T-cell depletion in a spontaneous metastasis model, we performed these experiments in EMT6 tumor-bearing mice. We treated EMT6-Luc tumor-bearing mice (at 4 weeks post implantation) with one dose of an anti-CD8α antibody or isotype control, after which the primary tumors were resected, and 2 days later, the second dose of an antibody was administered. Depletion of CD8$^+$ T cells in EMT6

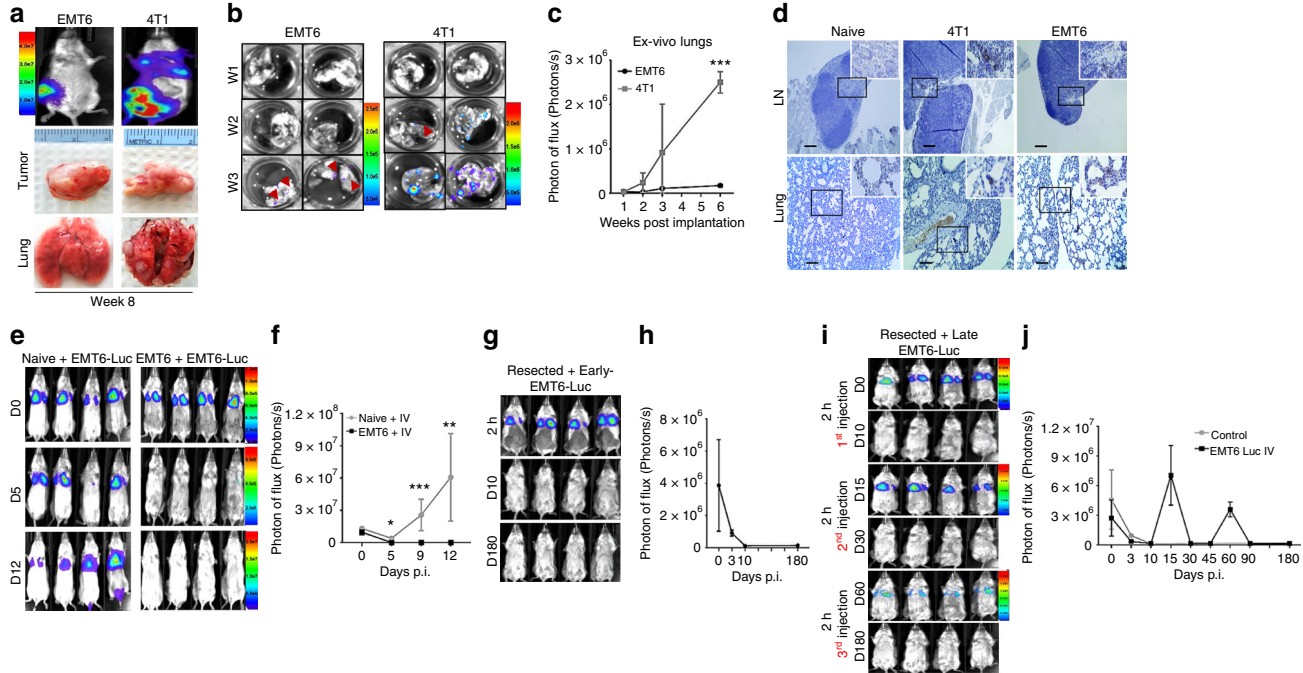

**Fig. 1** EMT6 tumor-bearing mice eradicate distant tumor cells. **a–c** EMT6 and 4T1 tumor cells (50K) establish similar size primary tumors in syngeneic mice, but only 4T1 tumor-bearing animals develop an overt metastasis. ***$P < 0.0005$, a two-tailed Student's $t$ test. **d** In situ analyses of tissues, using a pan-keratin antibody, reveal disseminated tumor cells in both 4T1 and EMT6 tumor-bearing mice. **e**, **f** Tail vein-injected EMT6-luc cells (100K) were eradicated in EMT6 tumor-bearing mice, while they established pulmonary metastasis in naive animals. Two-way analysis of variance test was used for comparisons. *$P < 0.05$, **$P < 0.005$, ***$P < 0.0005$. **g**, **h** Mice reject EMT6-Luc cells when injected via the tail vein 2 days post resection of primary EMT6 tumors. Proof of equally well-injected EMT6-Luc cells that were trapped in the lungs are shown 2 h post injection. **i**, **j** Mice 3 weeks post resection of primary EMT6 tumors, reject EMT6-Luc tumor cells following repeated injection via the tail vein. Results are presented as mean ± SD (3–10 mice were used for each group)

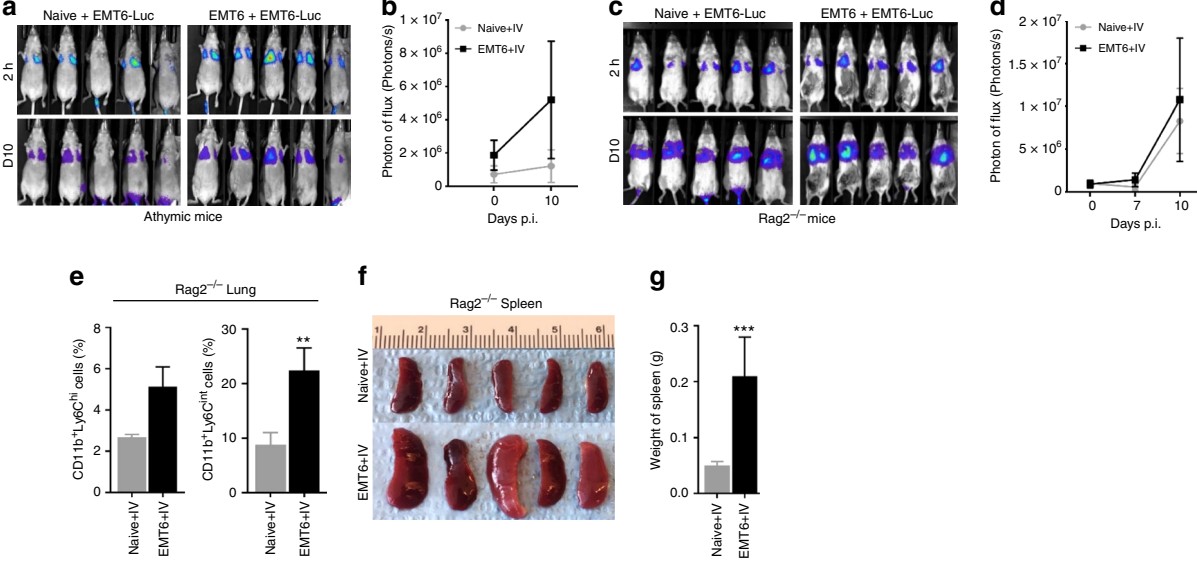

**Fig. 2** Eradication of distant tumor cells is T-cell dependent. **a–d** Athymic nude mice that lack T-cell development and Rag2$^{-/-}$ mice, deficient in mature T and B lymphocytes were implanted with or without EMT6 tumor cells (50K) and then challenged with EMT6-Luc cells (100K) via the tail vein injections at 3 weeks post implantation. Both tumor-bearing athymic and Rag2$^{-/-}$ mice, as well as their naive counterparts failed to eradicate distant tumor cells. Five animals were used for each group. **e–g** Tumor-bearing Rag2$^{-/-}$ mice exhibit moderately higher pulmonary metastasis, pulmonary gMDSC infiltration, and larger spleen size and weight. Results are presented as mean ± SD ($n = 3$). Two-tailed Student's $t$ test was used to compare groups. **$P < 0.005$, ***$P < 0.0005$

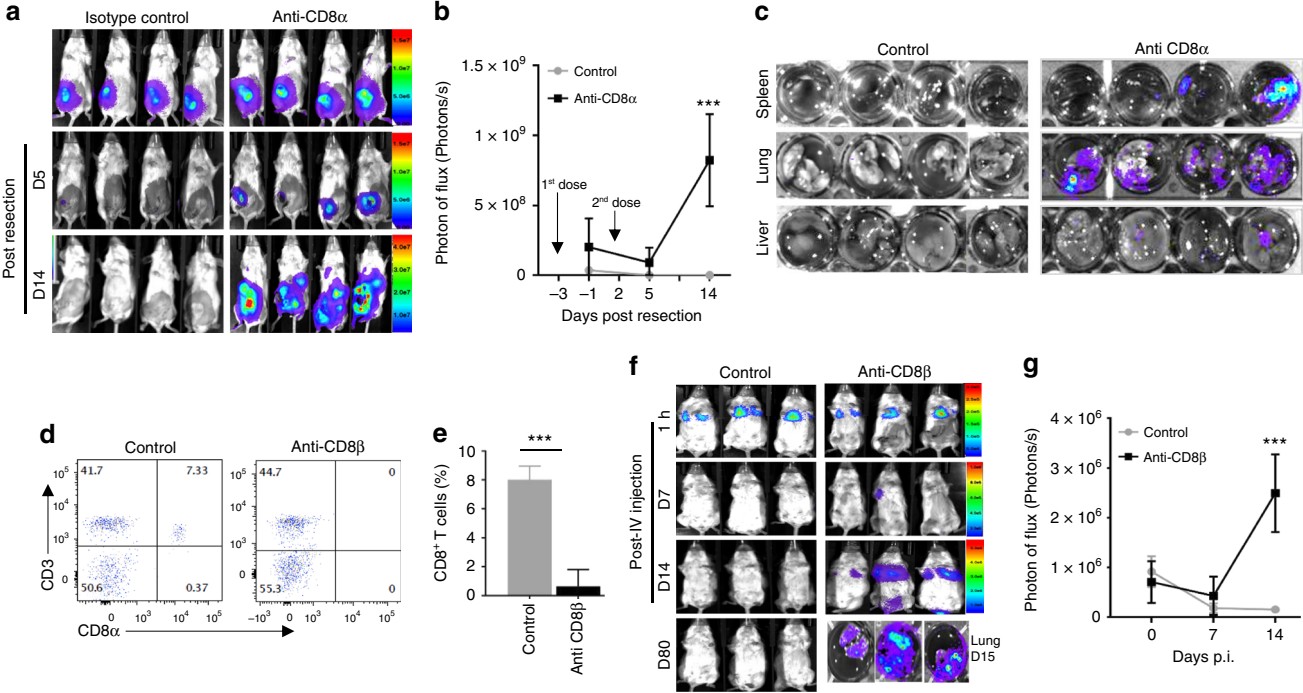

**Fig. 3** CD8+ T-cell depletion in EMT6 tumor-primed mice triggers the growth of DTCs. EMT6-Luc tumor-bearing mice at 4 weeks post implantation were treated with two doses of anti-CD8α antibody (500 μg/mouse); first dose 1 day before the resection of primary tumors and the second dose was administered 2 days after the resection. ***$P < 0.0005$, two-way analysis of variance test. **a–c** Depletion of CD8+ T cells enhanced the outgrowth of spontaneously disseminated tumor cells, when compared with isotype-treated mice, which showed no metastasis after the resection of primary tumors. EMT6-primed mice were treated with two doses of a CD8β antibody (500 μg/mouse); one dose 1 day before the tail vein injection of EMT6-Luc cells and the second dose 2 days after was administered. **d, e** CD8+ T cells were efficiently depleted in antibody-treated mice compared with the isotype control-treated animals. ***$P < 0.0005$, two-tailed Student's *t* test. **f, g** Depletion of CD8+ T cells promoted the outgrowth of tail vein-injected EMT6-Luc cells, while isotype-treated mice effectively eliminated these cells. Results are presented as mean ± SD ($n = 5$ for each group). ***$P < 0.0005$, two-way analysis of variance test

tumor-bearing mice resulted in increased local and distant metastasis, as demonstrated by live images of animals (Fig. 3a, b), as well as ex vivo images of lungs, livers, and spleens showed significantly higher luciferase activity in CD8+ T-cell-depleted animals (Fig. 3c).

We also examined this in the experimental metastasis model using EMT6-primed mice to provide more direct evidence. EMT6-primed mice were treated with an anti-CD8α antibody or isotype control 1 day before i.v. injection of EMT6-Luc cells, and 2 days after, a second dose of an antibody was administered. Pulmonary metastases were monitored by optical imaging, verifying that EMT6-Luc cells were trapped in the lungs after 1 h (Supplementary Fig. 3a). Mice treated with an isotype antibody effectively cleared the EMT6-luc cells (day 12) and remained tumor-free (day 21), whereas anti-CD8α antibody-treated mice developed pulmonary metastasis and died within 3 weeks (Supplementary Fig. 3a, b). Depletion of CD8+ T cells in antibody-treated mice was confirmed by flow-cytometry analyses (Supplementary Fig. 3c).

Because CD8α is also expressed on the surface of some natural killer (NK) cell subsets and dendritic cells[28], the use of an anti-CD8α antibody raises the possibility of NK cell involvement in tumor clearance. To distinguish the involvement of CD8+ T lymphocytes from that of NK cells, we utilized an anti-CD8β antibody for the depletion of CD8+ T lymphocytes that were effectively eliminated (Fig. 3d, e). Administration of the anti-CD8β antibody in EMT6-primed mice resulted in pulmonary metastasis within 2 weeks, while control isotype-treated mice cleared the tail vein-injected EMT6-Luc cells and remained tumor-free up to day 80 (Fig. 3f, g).

Furthermore, we assessed whether NK cells may also contribute to tumor-cell clearance, as previously reported[26,29,30]. Administration of an anti-asialo GM1 antibody significantly depleted NK cells in EMT6-primed mice compared with the control isotype-treated mice (Supplementary Fig. 3d). Expectedly, NK cell-depleted mice cleared pulmonary EMT6-Luc cells as effectively as the control isotype-treated animals (Supplementary Fig. 3e, f). Therefore, these results suggest that the antitumor response elicited by primary tumors in EMT6-primed mice is primarily CD8+ T-cell dependent and does not rely to a great extent on NK cells.

**EMT6 tumor-primed mice clear tail vein-injected tumor cells.** To examine in situ clearance of EMT6-luc cells in the lungs of EMT6-primed or naive mice, we evaluated the presence of tumor cells in lungs 2 h and 3 days post i.v. injection of EMT6-Luc cells. As a control, we included 4T1-primed mice that received EMT6-Luc cells. At 2 h post injection, EMT6-Luc cells were equally well accumulated in the lungs of naive, EMT6, and 4T1 tumor-bearing mice (Fig. 4a, b). At day 3 post injection, the EMT6 tumor-primed mice cleared the majority of the EMT6-Luc cells in the lungs (Fig. 4c, d). In contrast, the naive and 4T1 tumor-bearing mice displayed significantly higher luciferase signals of EMT6-Luc cells in lungs at the 3-day time point albeit with reduced Luci signal intensity compared with 2-h time points (Fig. 4c, d). Lungs from EMT6 tumor-bearing mice at day 3 post injection showed a low numbers of pan-keratin-positive cells, which correlated with greater proportion of apoptotic cells as quantified by the TUNEL assay (Fig. 4e, f). In contrast, lungs from naive and 4T1 tumor-bearing mice at the same time point exhibited a higher numbers

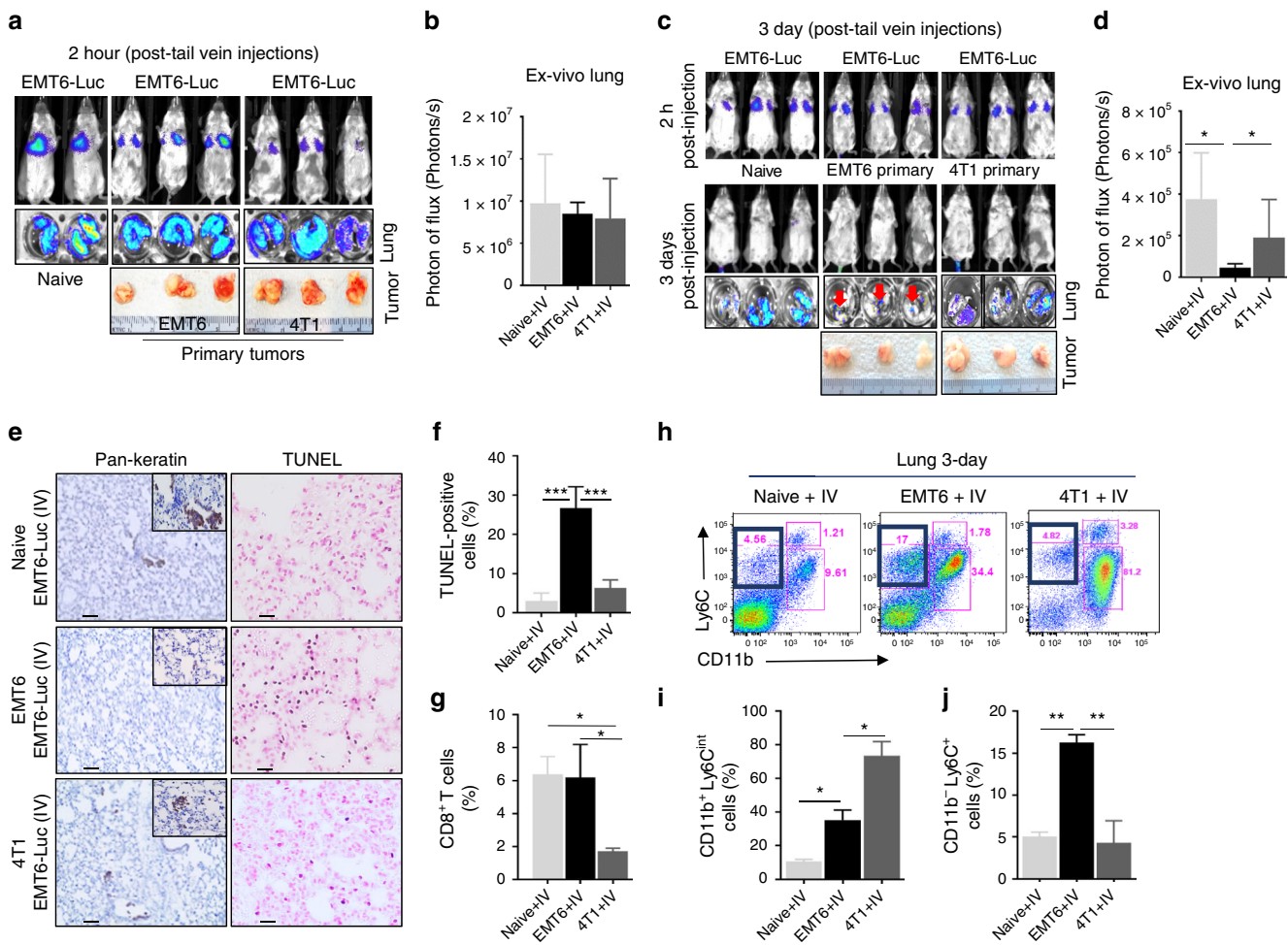

**Fig. 4** In situ clearance of EMT6-Luc cells in EMT6 tumor-primed mice. Primary tumor-bearing (EMT6 or 4T1) or the naive BALB/c mice were injected with EMT6-Luc cells (100K) via the tail vein 3-weeks post implantation. **a**, **b** Live animal and lung ex vivo bioluminescent images show entrapped EMT6-Luc cells after 2 h of tail vein injection in naive, EMT6, and 4T1 tumor-bearing mice, respectively. **c**, **d** Live imaging of mice and ex vivo imaging of lungs at 3 days post injection of indicated animals. The reduced signal intensity of EMT6-Luc cells in the lungs of EMT6 tumor-bearing mice compared with naive and 4T1 tumor-bearing mice is associated with less number of tumor cells. **e**, **f** Pan-keratin staining of tumor cells and TUNEL-positive apoptotic cells in the lungs of the indicated animals at day 3. **g** Flow-cytometry analyses of CD8$^+$ T cells in the lungs of naive, EMT6, and 4T1 tumor-bearing mice. **h**, **i** Flow-cytometry analysis shows an elevated CD11b$^+$Ly6C$^{int}$ cell population (gMDSC) in the lungs of 4T1 tumor-bearing mice compared with naive and EMT6 tumor-bearing mice. **j** The CD11b$^-$Ly6C$^+$ population is significantly increased in only EMT6 tumor-bearing mice at day 3 post i.v. injection. Results are presented as mean ± SD ($n = 3$). Scale bar: 50 μm. *$P < 0.05$, **$P < 0.005$, ***$P < 0.0005$, one-way analysis of variance test

of pan-keratin-positive cells and lower proportion of apoptotic cells (Fig. 4e, f). Consistent with the reduced tumor cells in lungs, higher levels of CD8$^+$ T cells were observed in the lungs of naive and EMT6-tumor-bearing mice as compared with the 4T1 mice (Fig. 4g). Also, lungs from 4T1 tumor-bearing mice were infiltrated with significantly higher levels of gMDSCs compared with the naive and EMT6 tumor-bearing mice (Fig. 4h, i). Interestingly, there was a significantly higher proportion of CD11b$^-$Ly6C$^+$ cells in the lungs of EMT6 tumor-bearing mice compared with the other groups (Fig. 4h–j), which could well represent Ly6C$^+$ effector CD8$^+$ T cells[31,32]. To provide further evidence of Ly6C$^+$CD8$^+$ T cells in EMT6-primed mice, we performed in situ analyses of axillary lymph nodes. Immunofluorescent staining showed a higher infiltration of Ly6C$^+$CD8$^+$ T cells in lymph nodes from EMT6-primed mice compared with those from naive and 4T1 tumor-bearing animals (Supplementary Fig. 4).

**Ly6C$^+$CD8$^+$ subset exhibits effector T-cell function.** We previously reported that inefficient MDSC induction in EMT6

tumor-bearing mice may account for its failure to generate spontaneous metastasis[24]. To determine whether CD8$^+$ T-cell depletion had any impact on MDSC population, we performed flow-cytometry analyses of PBMCs collected from naive, isotype control and anti-CD8 antibody-treated mice. There was no statistically significant difference in MDSC populations between the isotype control and anti-CD8 antibody-treated mice (Fig. 5a). However, we observed that the CD11b$^-$Ly6C$^+$ cell population was substantially reduced in the T-cell-depleted mice (Fig. 5a–c). It is possible that the CD11b$^-$Ly6C$^+$ cells observed in EMT6-primed mice (Fig. 4h, i) could be effector/memory CD8$^+$ T cells that may be involved in antitumor immunity. Therefore, we examined whether the CD11b$^-$Ly6C$^+$ cell population expressed the CD8 CTL marker and analyzed whether anti-CD8 antibody treatment depleted this population. Indeed, we observed that CD11b$^-$Ly6C$^+$ cell population also expresses a CD8 marker (Fig. 5c) and anti-CD8 antibody treatment depleted these cells (Fig. 5d, e).

Next, we examined the cytotoxic activity of Ly6C$^+$CD8$^+$ or Ly6C$^-$CD8$^+$ T-cell populations isolated from the spleens of

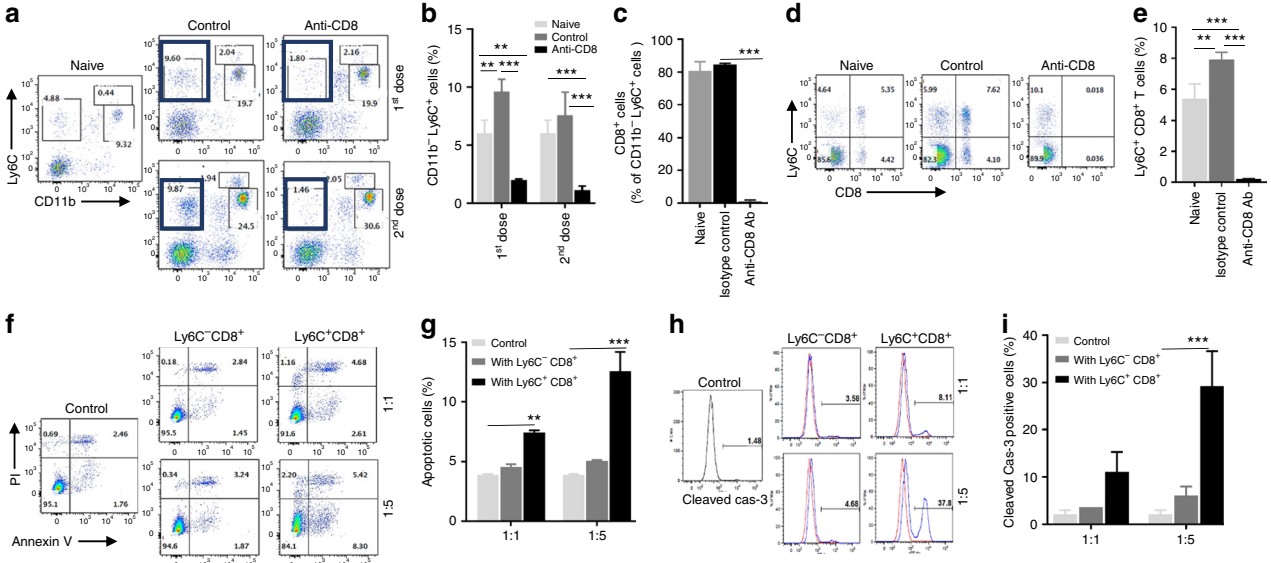

**Fig. 5** Ly6C⁺CD8⁺ T-cell population exhibits effector function. EMT6-primed were injected with 2 doses of CD8 depletion antibody 6 months after removal of primary tumors. **a, b** A significant reduction in the circulating CD11b⁻Ly6C⁺ population was observed in CD8⁺ T-cell depleted mice compared with control and naive animals. **c–e** Treatment with anti-CD8α antibody depleted more than 80% of CD8⁺ T cells, majority of which express Ly6C surface marker. **f, g** Tumor cell killing assay was performed to determine the effector activity of Ly6C⁺CD8⁺ or Ly6C⁻CD8⁺ subsets. The spleen-derived Ly6C⁺CD8⁺ subset displays much higher tumor-cell killing activity than the Ly6C⁻CD8⁺ subset against EMT6 tumor cells as demonstared by Annexin V staining of tumor cells. **h, i** The Ly6C⁺CD8⁺ subset also induced more cleaved-caspase expression in EMT6 tumor cells when co-ultured 1:5 (Tumor: T cells) ratio. Results are presented as mean ± SD ($n = 3$). *$P < 0.05$, **$P < 0.005$, ***$P < 0.0005$, one-way analysis of variance test

EMT6 tumor-primed mice in vitro. When cocultured with EMT6 tumor cells, Ly6C⁺CD8⁺ T-cell population showed significantly higher tumor cell killing activity compared with the Ly6C⁻CD8⁺ subset as demonstrated by increased annexin V staining and caspase 3 activity in the target tumor cells (Fig. 5f–i). Identical T-cell populations isolated from naive (non-tumor-bearing) mice failed to show cytotoxic activity against EMT6 tumor cells (Supplementary Fig. 5a, b).

**Gene signature predicts favorable survival in TNBC patients.** Having established that the Ly6C⁺CD8⁺ T-cell population exhibits cytotoxic activity, we examined the expression of genes related to effector function in this population. Consistent with the cytotoxic activity, Ly6C⁺CD8⁺ population from EMT6 tumor-primed mice expressed higher levels of *IFNG, GZMA, GZMB,* and *PRF1* as assessed by qPCR analyses (Fig. 6a). Due to the fact that more than 80% of the Ly6C⁺ cell population was CD8 positive, we performed microarray analyses of lung-derived Ly6C⁺ cell populations (gated on the CD45⁺) from 4T1 and EMT6 tumor-bearing mice at 3 weeks post implantation. The microarray data revealed 728 distinctly regulated genes between two populations and the top highly expressed genes (*IL2RB, GZMA, GZMB, EMR4, PRF1, CX3CR1, STAT1,* and *TLR9*) in lung-derived Ly6C⁺ cells from EMT6 tumor-bearing mice are associated with effector T-cell function (Fig. 6b, c and Supplementary Data 1). We hypothesized that EMT6 tumor-bearing mice may have elicited an early antitumor immune response against the primary tumors, which express an immune-activated gene signature at early time points. To demonstrate this, we examined the expression of these genes in primary tumors at 1, 2, and 3 weeks post implantation. Although there was no significant difference in the expression of this signature genes in primary tumors isolated between 1 and 2 weeks post implantation, they were significantly suppressed by 3 weeks post implantation (Fig. 6d), suggesting either limited recruitment of tumor-infiltrating lymphocytes (TILs)[33] or gradual suppression of cytotoxic T lymphocytes. This

is indeed consistent with the rapid growth of primary EMT6 tumors between 2 and 3 weeks post implantation. Therefore, our findings prompted us to test whether the expression of this immune signature in breast cancer patients can predict favorable overall survival. Using the TCGA data set[34], we found that breast cancer patients whose tumors expressed higher levels of immune signature genes had significantly better overall survival (Fig. 6e). Interestingly, all of the patients with triple-negative breast cancer expressing one or more of the immune signature genes survived during the course of TCGA data collection (Fig. 6f). However, this signature did not predict better overall survival in HER2 + and luminal breast cancer patients (Fig. 6g, h). In line with favorable overall survival, the immune activation gene signature also positively correlated with higher CD8 gene expression in TNBC patients (Fig. 6i).

**Mice eliminate DTCs after complete resection of primary tumors.** Although EMT6 tumor-primed mice clear DTCs or the tail vein-injected cells via a CD8⁺ T-cell dependent manner, these results could also be explained by the existence of dormant tumor cells. Since DTCs generated metastasis when CD8⁺ T cells were depleted around the time of resection of primary tumors (Fig. 3d–f), we reasoned that if there are disseminated solitary tumor cells in a dormant state, they will be activated following the depletion of CD8⁺ T cells long after the resection of primary tumors. In this model, we depleted CD8⁺ T cells 60 days after the resection of primary EMT6-Luc tumors, and monitored the mice by optical imaging for up to 150 days (Fig. 7a).

Although expectedly, isotype-treated control mice were free of metastasis up to 150 days post resection of primary tumors, anti-CD8 antibody-treated mice did not show any luciferase signal during the same time (Fig. 7b). We confirmed an efficient CD8⁺ T-cell depletion as demonstrated by analyses of peripheral blood mononuclear cells (PBMC) collected via tail vein during the course of 4 weeks post treatment with an anti-CD8 antibody (Fig. 7c, d). To further determine whether there are solitary

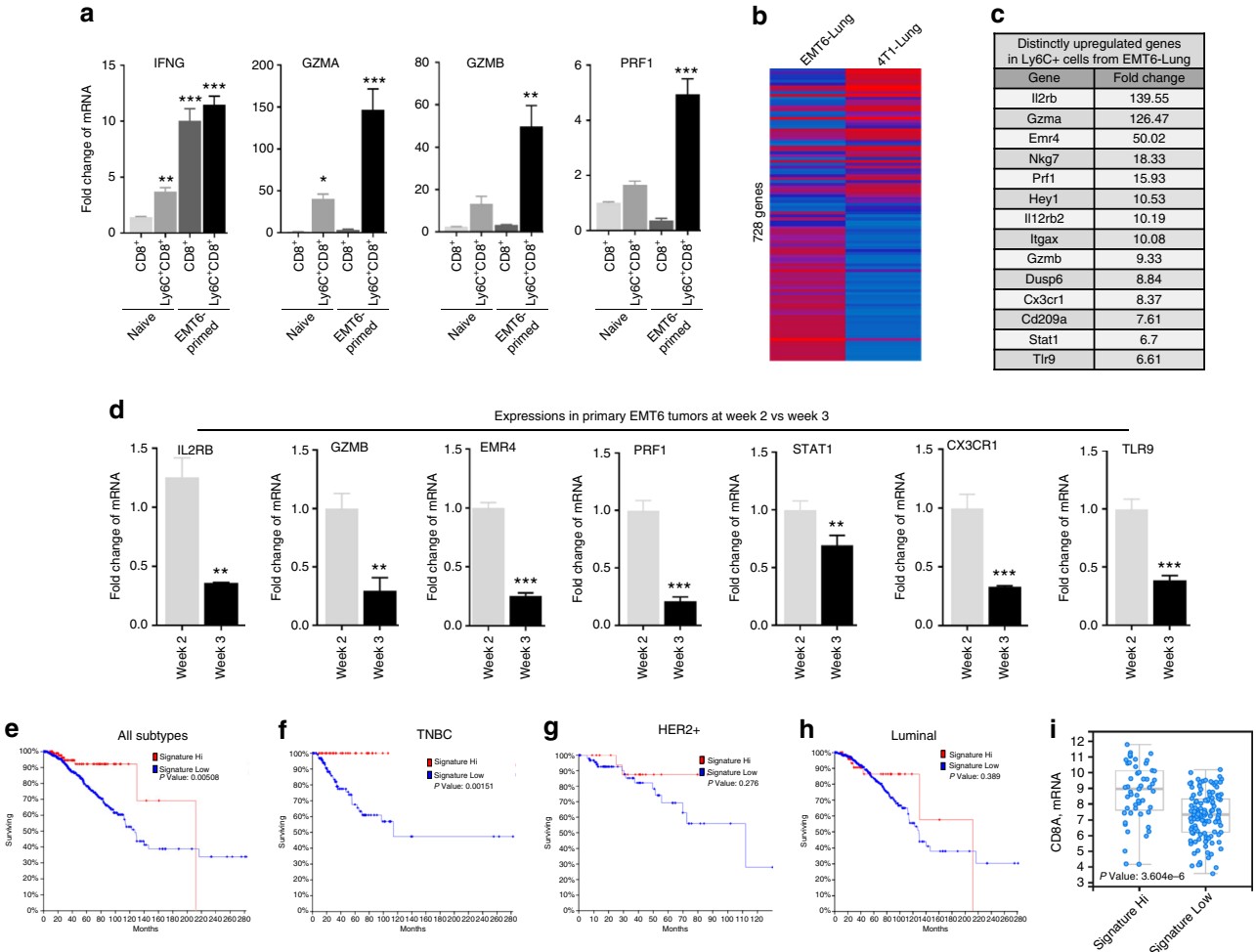

**Fig. 6** Immune activation gene signature predicts better overall survival in breast cancer patients. **a** Expressions of effector T-cell-specific genes were higher in Ly6C+CD8+ population compared with the Ly6C−CD8+ subset isolated from the spleen of EMT6 tumor-bearing mice and counterparts isolated from naive mice. *$P < 0.05$, **$P < 0.005$, ***$P < 0.0005$, one-way analysis of variance test. **b, c** Gene expression analyses of lung-derived Ly6C+ cell population from EMT6 tumor-bearing mice compared with naive or 4T1 tumor-bearing mice revealed 728 distinctly regulated genes. Top highly expressed genes are related to T-cell activation called "immune activated gene signature". **d** Expressions of these immune-activated genes in primary EMT6 tumors at early time points suggest that early immune response. **$P < 0.005$, ***$P < 0.0005$, two-tailed Student's $t$ test. Results are presented as mean ± SD ($n = 3$). **e–h** Higher expression of immune-activated gene signature predicts better overall survival in breast cancer patients. Patients with TNBC subtype derive the highest survival benefit, while Luminal and HER2 + patients fail to show any benefit. **i** Elevated expression of immune-activated gene signature correlates with higher expression of CD8 mRNA in TNBC tumor samples

dormant tumor cells in the lungs, mice were killed at 150 day post resection, and lung sections were subjected to IHC analyses using a pan-keratin antibody. However, we failed to detect any pan-keratin positive cells in the lungs of animals that are treated with an anti-CD8 antibody (Fig. 7e).

**Residual tumors enhance tumor growth and metastasis**. Although the complete resection of primary EMT6 tumors cures mice, we were intrigued by the rapid progression of tumors in mice with residual tumors following the resection of primary tumors. Therefore, we set out to examine the differences in these two sets of mice; one with complete resection and other with residual tumors following the surgical removal of primary tumors. A recent report by Weinberg and colleagues suggested that systemic inflammation to surgery triggers the outgrowth of distant immune-controlled tumors[18]. Consistent with this report, mice with residual tumors after the surgery exhibited enhanced growth of relapsed tumors and promoted the outgrowth of DTCs (Fig. 8a–d). On the basis of our previous report[24], we examined

the gMDSC population in the lungs and spleens of mice 3 weeks post resection. Mice with no residual tumors showed a modest gMDSCs infiltration in the lungs and spleens (Fig. 8e, f), which is similar to what we found in naive mice[24]. In contrast, mice with residual tumors exhibited up to 4–10-fold higher infiltration of gMDSCs in their lungs and spleens (Fig. 8e, f). Consistent with the preceding experiments, we also found that CD11b−Ly6C+ population was significantly depleted in mice with residual tumors (Fig. 8e, g). Because the majority of CD11b−Ly6C+ population are CD8+ T cells, we next evaluated the Ly6C+CD8+ T-cell population in these organs. Expectedly, mice with complete resection of primary tumors sustained a substantial Ly6C+CD8+ population in the spleens and lungs, whereas mice with residual tumors were deprived of this population in their respective organs (Fig. 8h). We previously reported that inflammatory cytokines such as G-CSF and IL-6 may generate a systemic permissive microenvironment conducive to the establishment of metastasis[24,35,36]. Krall et al. corroborated this by demonstrating an upregulation of inflammatory cytokines, G-CSF, and IL6 in

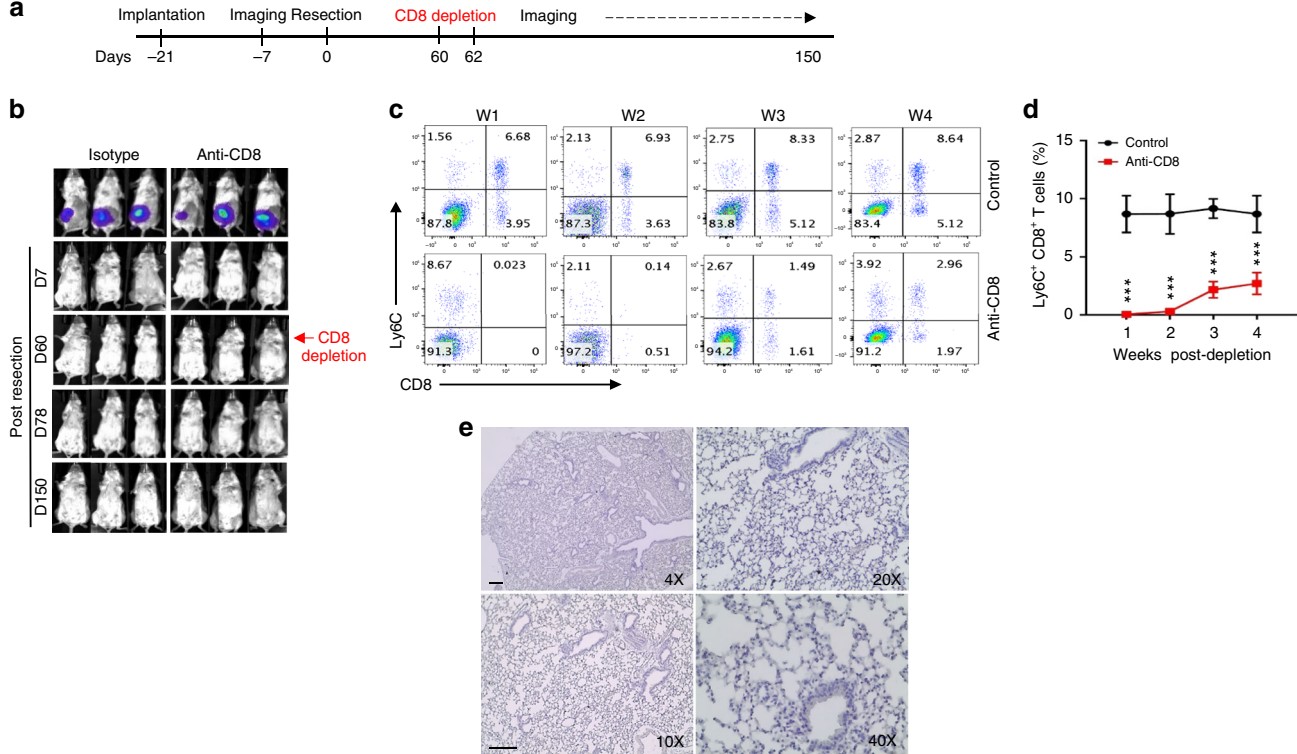

**Fig. 7** DTCs are eliminated after the complete resection of primary EMT6 tumors. **a** Time line of experimental procedures. **b** EMT6-Luc cells were first implanted orthotopically into the mammary fat pads of BALB/c mice ($n = 5$–10 for each group), primary tumors were resected 3 weeks post implantation, and the anti-CD8 antibody treatment was started 60-day post resection. Animals were then monitored by weekly bioluminescence imaging up to 150 days and no signal was detected in both control- and antibody-treated groups. **c**, **d** CD8$^+$ T-cell depletion in treated animals was monitored weekly for 4 weeks post treatment and a significant depletion of CD8$^+$ T cells was demonstrated by analyzing the PBMCs. **e** Mice were killed and pan-keratin staining was performed in the lung sections of treated animals showing no detectable pan-keratin positive cells. Scale bar: 50 μm. Results are presented as mean ± SD ($n = 3$). ***$P < 0.0005$, two-way analysis of variance test

response to surgery in animals with residual tumors[18]. Consistent with their findings, we show a several 1000-fold higher expressions of G-CSF and IL6 in relapsed tumors post surgery as compared with their expression in primary EMT6 tumors (Fig. 8i). Furthermore, we and others demonstrated that G-CSF stimulates the induction and mobilization of MDSCs from the bone marrow[24,37,38]. Due to the fact these findings suggest a critical role for G-CSF in generating a metastatic permissive microenvironment, we tested whether treatment of EMT6-primed mice with recombinant G-CSF would promote metastasis. Treatment of EMT6-primed mice with recombinant G-CSF (daily for 2 weeks) effectively induced gMDSC accumulation and diminished the Ly6C$^+$CD8$^+$ T-cell population (Fig. 8j–m), resulting in increased metastatic growth of DTCs in these animals (Fig. 8n, o, and q).

We previously reported that a granulocytic subset of gMDSC plays an important role in immune suppression and promoting pulmonary metastasis in the 4T1 tumor model[24]. After having established that the clearance of DTCs is mediated by CD8$^+$ CTLs, we determined whether gMDSCs from 4T1 tumor-bearing mice could suppress the CTL activity in EMT6 tumor-primed model allowing the establishment of lung metastasis. Therefore, gMDSCs (250,000 cell dose derived from the lungs of 4T1 tumor-bearing mice) were injected into EMT6 tumor-primed mice 1 day before the i.v. injection of EMT6-Luc cells and a second gMDSC dose 5 days later. Mice receiving gMDSCs developed pulmonary metastasis within 2 weeks while the control animals did not (Fig. 8p, q). All together these experiments suggest that EMT6 primary tumors generate anti-metastatic CD8$^+$ T-cell responses,

and that gMDSCs derived from 4T1 tumor-bearing mice can inhibit this antitumor activity allowing the tumor cells to establish secondary metastasis (Fig. 9).

## Discussion

Although it has long been appreciated that dissemination of tumor cells may occur during early stages of carcinogenesis, the role of these early disseminated cells in forming clinically significant metastasis has remained unclear[1,3,10,39]. Here, we provide evidence that primary tumor-induced immune response clears DTCs following the complete resection of primary tumors in syngeneic mouse model. Our findings support the concept that DTCs may be eliminated by the innate and adaptive immune surveillance[3,9]. Although the effect of surgical resection of primary tumor on DTCs is debated, recent clinical evidence suggests that the complete resection of primary breast tumors with negative surgical margins significantly improves survival of patients[14,40]. This may also be true for other solid malignancies with clear resectable margins[41,42]. In order to develop a mouse model to investigate the fate of DTCs, we utilized two murine tumors: 4T1, an extremely metastatic in syngeneic BALB/c model[24] representing the triple-negative breast cancer subtype[43] and EMT6, a less invasive weakly metastatic in syngeneic BALB/c mice[44,45]. We first established that DTCs are detectable as early as 1 week post implantation in draining lymph nodes and lungs in both 4T1 and EMT6 tumor models. However, EMT6 tumor-bearing mice fail to develop readily detectable macrometastasis despite the existence of DTCs, while 4T1 tumor-bearing mice

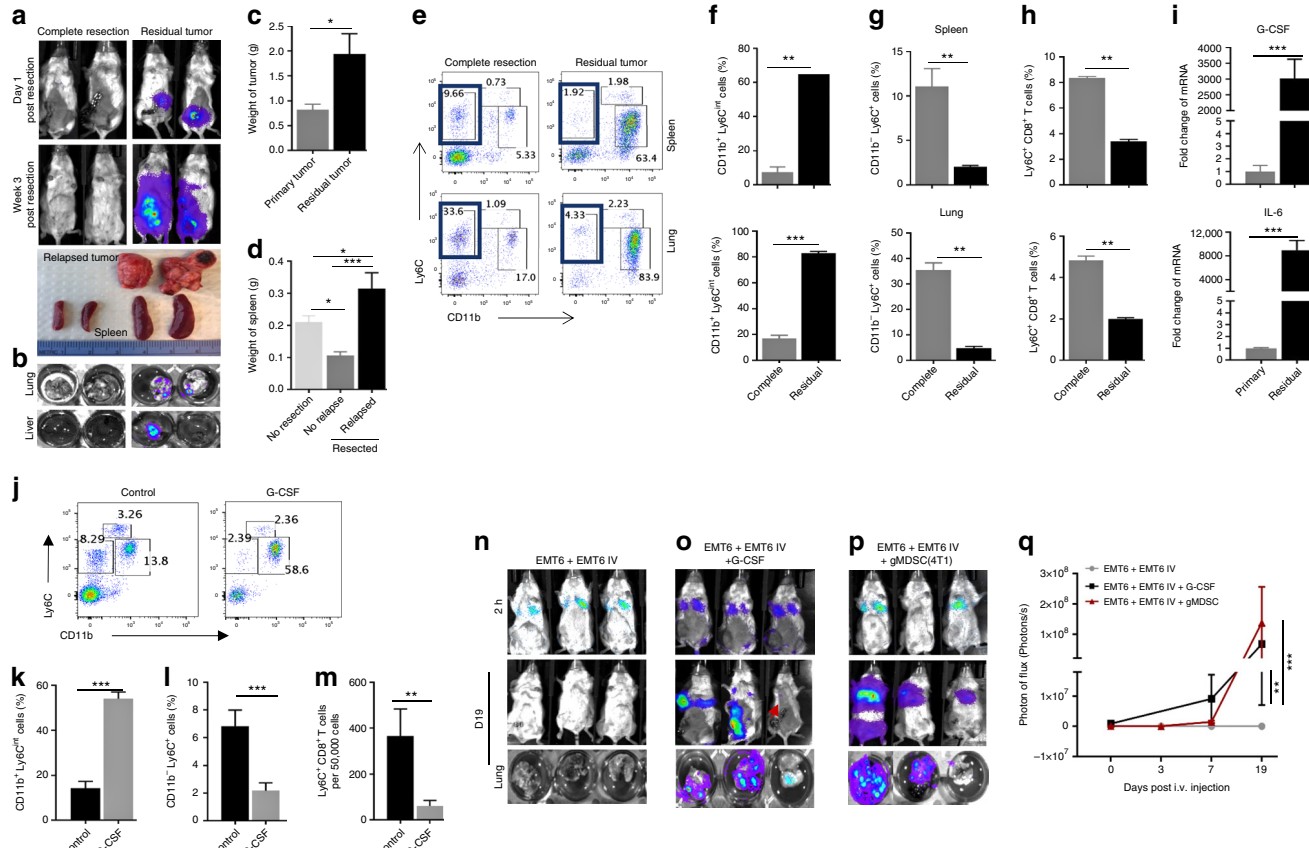

**Fig. 8** Residual tumors promote local tumor growth and the outgrowth of DTCs. **a–d** Mice with residual tumors following surgical resection show a rapid growth of relapsed tumors, enlarged spleen, and promotes metastatic outgrowths compared with mice without residual tumors or primary tumor-bearing mice. *$P < 0.05$, ***$P < 0.0005$, one-way analysis of variance test. **e, f** Granulocytic MDSC population defined by CD11b+Ly6C$^{int}$ phenotype is significantly expanded. **g, h** Ly6C+CD8+ T-cell subset out of CD11b−Ly6C+ population is significantly reduced in the spleens and lungs of mice with residual tumors compared with those from tumor-free mice. **i** G-CSF and IL6 expressions are increased by several 1000-fold in residual tumors compared with the primary tumor. **j–m** Administration of recombinant G-CSF (40 μg/mouse) in EMT6-primed mice induced the gMDSC accumulation and suppression of effector T cells with CD8 + Ly6C + phenotype. **$P < 0.005$, ***$P < 0.0005$, two-tailed Student's $t$ test. **n–q** Administration of recombinant G-CSF or adoptive transfer of gMDSCs from 4T1 tumor-bearing mice (100K cells injected twice via tail vein) promotes the growth of distant tumor cells. Results are presented as mean ± SD ($n = 3$). **$P < 0.005$, ***$P < 0.0005$, two-way analysis of variance test

exhibit progressive metastasis as previously demonstrated by us and others[24,26].

Tumor metastasis is an inherently inefficient process, with only a small fraction of DTCs capable of metastatic colonization while the majority of tumor cells are eliminated either in the circulation or in secondary organs[46,47]. Due to its inefficiency of establishing metastasis despite the DTCs, we considered that EMT6 tumor-bearing mouse would be a suitable model to investigate the fate of DTCs. Specifically, we asked the question whether the DTCs are cleared by the immune system[3,9] or maintained in a dormant state[6–8]. The complete clearance of EMT6-Luc cells (100K) in the lungs of EMT6 tumor-bearing mice suggested that the primary EMT6 tumor induces antitumor immunity by which it protects animals from developing metastasis. We demonstrated that this immunity was CD8+ T-cell dependent, because primary tumors in athymic nude or Rag2$^{−/−}$ mice, lacking functional T cells, did not protect animals from developing metastasis. We confirmed this assumption and extended these findings in syngeneic BALB/c model where depletion of CD8+ T cells resulted in metastatic growth in both experimental and spontaneous metastasis models. In keeping with our findings, depletion of CD8+ T cells resulted in metastatic outgrowths in the spontaneous mouse melanoma model[47]. It was also reported that the innate immune surveillance, including NK cells elicited by the primary tumor may limit

the formation of metastasis in the melanoma model[29,30,48]. However, we determined that the tumor cell clearance in EMT6-primed mouse model was primarily mediated by CD8+ CTLs and not dependent on NK cells.

Having established that EMT6 primary tumor may elicit CD8+ T-cell-dependent antitumor response to limit metastatic outgrowths, we considered that it may be an ideal platform to determine the fate of DTCs following the resection of primary tumors. Mice with the complete resection of primary tumors not only developed metastasis for up to 6 months, but also rejected EMT6-Luc cells when they were repeatedly injected via tail vein. The fact that animals were able to reject the tail vein-injected EMT6-Luc cells even after 4 weeks post resection of primary tumors, suggested that an immunological memory was formed. This is in line with previous reports that widely established the existence of memory T cells in animals that reject tumors upon repeated challenge[49,50]. Collectively, our research supports the notion that immune surveillance may limit metastatic outgrowth of DTCs[3,9,51].

Analyses of PBMCs in T-cell-depleted mice with EMT6 tumors revealed a substantial decrease of the CD11b−Ly6C+ population. Consistent with previous reports that effector and memory CD8+ T cells express the Ly6C surface marker[31–33], we show that more than 80% of CD11b−Ly6C+ population were indeed CD8+

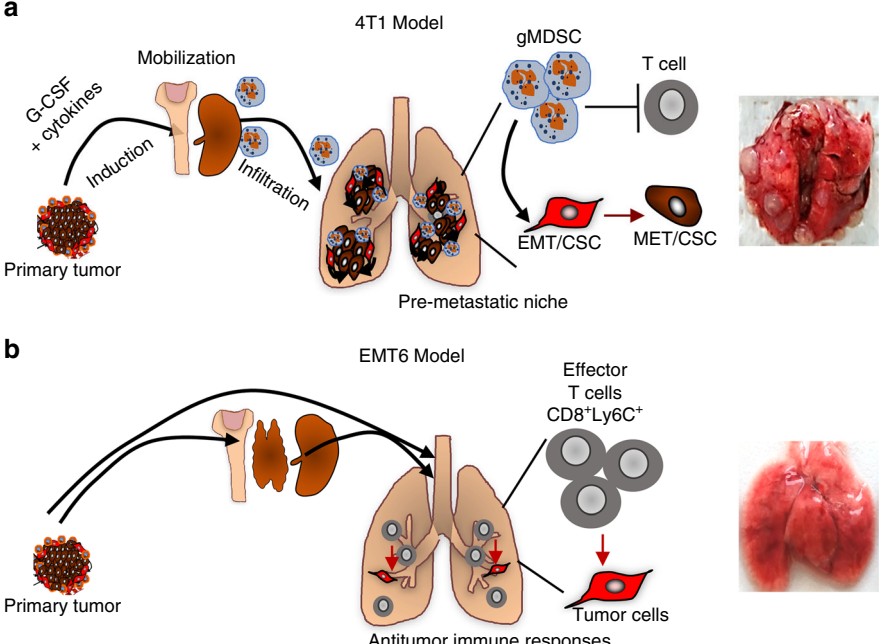

**Fig. 9** Illustration of key differences between the metastatic and non-invasive tumors. **a** The 4T1 tumor model effectively induces immunosuppressive MDCS accumulation by secretion of inflammatory cytokines such as G-CSF and that generates permissive pro-metastatic microenvironment. **b** The EMT6 tumor model, however, fails to induce MDSC accumulation and resulting in antitumor immune response that effectively eradicates disseminated tumor cells in distant organs

T cells. Moreover, Ly6C$^+$CD8$^+$ population from EMT6 tumor-primed mice showed much higher effector activity compared with Ly6C$^-$CD8$^+$ population, suggesting that this CD8$^+$ T-cell population involved in eradication of DTCs. Furthermore, gene expression analyses of Ly6C$^+$ cell populations revealed a higher expression of immune activation genes in the lungs of EMT6 tumor-bearing mice at 3 weeks post implantation compared with the ones from 4T1 tumor-bearing mice. These data further supported our findings that EMT6 tumor-bearing mice exhibit an antitumor immunity in distant organs. Interestingly, recent clinical data suggested that the presence of TIL predict improved overall survival in all subtypes of breast cancer, however, patients with TNBC subtype exhibit a higher level of TILs and drive the most survival benefit compared with other subtypes[52–54]. In line with these clinical data, our immune activation gene signature predicted better overall survival in TNBC subtype compared with other subtypes.

Although only a few studies investigated the effect of surgery on DTCs using various mouse models[17,18,20], none have addressed the fate of DTCs after complete resection of primary tumors. In our model, the fact that mice do not develop any metastasis following the complete resection of primary tumors suggested two likely scenario; one is that DTCs are cleared by antitumor immunity[3,9,51] or they are maintianed in a dormant state[6–8]. Because the depletion of CD8$^+$ T cells 60 days post resection of primary EMT6 tumor cells failed to induce any metastatic outgrowth, we concluded that an immune eradication of DTCs is the most likely scenario in our model. In addition, the lack of any pan-keratin positive tumor cells in the lungs of these animals provided additional support for the eradication of DTCs.

In contrast to complete resection, mice with residual tumors following surgery exhibited an enhanced growth of local tumors and DTCs. Our results may be reconciled with the recent report by Krall et al.[18], which suggested that systemic inflammatory response to surgery promotes the outgrowth of distant immune-controlled tumors in the mouse dormancy model.

Interestingly, mice with residual tumors exhibited significantly higher gMDSC infiltration (more than 60%) in the lungs and spleen compared with the mice with complete resection of tumors which showed between 7 and 18% gMDSC infiltration, respectively. Furthermore, Ly6C$^+$CD8$^+$ population was significantly diminished in mice with residual tumors, suggesting that higher levels of gMDSCs effectively suppressed the effector CD8$^+$ T-cell population.

We previously demonstrated in the 4T1 tumor model that gMDSCs play a critical role in metastatic outgrowth of DTCs[24]. Therefore, we reasoned that the failure of DTCs to establish metastatic outgrowths in the EMT6 tumor model may be due to its inefficiency to induce suppressive gMDSCs. Consistent with this hypothesis, we showed here that gMDSCs isolated from 4T1 tumor-bearing mice induced the metastatic outgrowth of EMT6-Luc cells when they were co-injected via tail vein. It has been established by us and others that gMDSCs are immature myeloid cells that suppress T-cell function via expression of mediators, such as arginase and iNOS[24,55–57]. Furthermore, G-CSF is known to induce the mobilization of MDCS from the bone marrow[24,37,38]. Consistent with this, we showed here that administration of G-CSF to EMT6-primed mice promoted the growth of DTCs by inducing gMDSCs that suppress Ly6C$^+$CD8$^+$ effector T cells.

In summary, our experiments provide a compelling evidence for the immune-mediated eradication of DTCs following the complete resection of primary tumors. In this model, eradication of DTCs is dependent on the effector Ly6C$^+$CD8$^+$ T-cell population as well as inability of tumor at inducing a suppressive gMDSC population. However, mice with recurrent tumors following surgery exhibit enhanced growth of relapsed primary tumors and outgrowth of DTCs. A massive gMDCS infiltration in the spleens and lungs of animals with residual tumor after surgery, perhaps consistent with the recently reported dormancy model where mobilization of myeloid cells in response to surgical wounding supported the growth of immune-controlled tumors[18].

Furthermore, our studies may also provide molecular explanation of improved overal survival in breast cancer patients following complete resection of primary tumors with negative margins[14].

## Methods

**Cell lines.** EMT6 and 4T1 cell lines were purchased from American Type Culture Collection (ATCC). All cell lines were analyzed for mycoplasma contamination using MycoAlert Mycoplasma Detection Kit (Lonza). All cell lines were maintained in RPMI supplemented with 10% fetal bovine serum, and antibiotic/antimycotic 10,000 units/ml. These cell lines were infected with Luciferase expressing lentivirus, and stable cell lines were generated. EMT6-Luc and 4T1-Luc cells were used to monitor tumor growth in live animals and tissues.

**Mouse tumor implantation.** All mice procedures were conducted in accordance with the Institutional Animal Care and Use Committee at Augusta University (AU). The animal protocol for the procedures conducted in this study is approved by the Laboratory of Animal Services (LAS) at AU. Five-week-old Balb/c female mice, Rag2$^{-/-}$ female mice, and athymic nude female mice were purchased from Charles River at NCI. As indicated in the figure legends, 3–5 mice in each group were used.

Parental or luciferase expressing tumor cells (EMT6 or 4T1) cells (in 50% matrigel) are surgically implanted into the mammary fat pads of anesthetized mice (Ketamine/Xylazine) by an L-shaped incision (1 × 1 cm) between the abdominal midline and the 4th and 5th nipples. Once the skin flap is opened with moist cotton swabs and the mammary fat pads are exposed, cells are injected into the fat pads with an insulin syringe in maximum 50 μl of volume. Skin flaps are sutured by using clips.

**Resection of primary tumors and tail vein injections.** Primary EMT6 tumors at 3–4 weeks post implantation were surgically resected by making incision around the tumor mass, flip open the skin covering the tumor and mammary fat pad. Once the primary tumor is exposed, arteries supplying the tumor were cauterized, and tumor as well as draining lymph nodes were carefully removed. Skin flaps were sutured by metal clips. In cases of primary tumors expressing luciferase, animals were imaged by bioluminescence to determine whether any residual tumors were left after surgery.

For tail vein injections, 100K EMT6-Luc tumor cells were suspended in 100 μl of saline solution. Animals were first kept under light to warm the tail veins and then transferred to restrainer where the tumor cells were injected via the tail vein. Animals were then imaged by bioluminescence 1–2 h after injection when the tumor cells trapped in the lungs. These animals were monitored weekly by bioluminescence imaging.

**T lymphocyte depletion.** The CD8$^+$ T lymphocyte depletion in indicated mice was performed by two injections of purified anti-CD8α antibody (Bio X Cell, clone 2.43) or anti-CD8β (Bio X Cell, clone 53–5.8) at the dose of 500 μg/mouse with 2 days interval between the doses. In the spontaneous metastasis model, first dose was introduced before resection and 2nd dose was injected 2 days post resection of primary tumor. Control mice were treated with a single i.p. injection of 500 μg of non-immune rat IgG (ICN Pharmaceuticals, Aurora, OH). The depletion was verified by flow-cytometry analyses of PBMCs following the treatment with anti-CD8 antibody.

**NK cell depletion.** To deplete NK cells, anti-asialo GM1 antibodies (BioLegend, clone Poly21460) at the dose of 20 μL per mouse, were injected i.p. 1 day before i.v. injection of EMT6-Luc cells into EMT6-primed mice. Efficient depletion of NK cells were shown by flow analysis of PMBCs 3 and 5 weeks post injection of depletion antibodies. Anti-Asialo GM1 (#146007, Biolegend) and NKp46 (# 137605, Biolegend) are used to determine the NK cells in CD3-negative lymphocytes population.

**In vivo G-CSF administration.** Recombinant G-CSF (Gemini) was injected (IP) daily at the dose of 40 μg/mouse for 2 weeks after the implantation of EMT6 tumor cells into the mammary fat pads of mice. One-week after the G-CSF injection, 1 × 10$^5$ EMT6-Luc cells were administered through the tail vein and primary tumor were resected at day 9. Recombinant G-CSF injections were continued one more week after i.v. injection of tumor cells. Induction and mobilization of gMDSCs (CD11b$^+$ Ly6C$^{int}$) were determined by flow analysis of PMBCs. Mice were followed by BLI for metastatic colonization of EMT6-Luc cells.

**RNA extraction and real-time RT–PCR.** The total RNA was extracted using RNeasy Mini kit (Qiagen), and 500 ng of RNA was used to make cDNA using iScript cDNA synthesis kit (Bio Rad). For RT–PCR analyses, following gene specific primers ordered from KiCqStart SYBR Green predesigned primers (Sigma) were used:

IFNG (F—5′-TGAGTATTGCCAAGTTTGAG-3′, R—5′-CTTATTGGGACAA TCTCTTCC-3′), GZMB (F—5′-CTGCTAAAGCTGAAGAGTAAG-3′, R—5′-

TAGCGTGTTTGAGTATTTGC-3′), GZMA (F—5′-CTTGCTACTCTCCTTTT TCTC-3′, R—5′-CTTAGATCTCTTTCCCACG-3′), PRF1 (F—5′-AATTTTGCAG CTGAGAAGAC-3′, R—5′-CTGTGGAGCTGTTAAAGTTG-3′), IL2RB (F—5′-A AGCTCAACGAAACAATACC-3′, R—5′-ACTTGACCAAATGTAGACG-3′), EM R4 (F—5′- AATATTCAGCCCATTGACTC-3′, R—5′-AACACTTGCAAATGGT GTAG-3′), STAT1 (F—5′-TTTGACAGTATGATGAGCAC-3′, R—5′-AGCAAAT GTGATGCTCTTTC-3′), CX3CR1 (F—5′-AACACCATGCTGTCATATTC-3′, R —5′-GTAAGCTACTATGCTTGCTG-3′), and TLR9 (F—5′-TCTCCCAACATG GTTCTC-3′, R—5′-CTTCAGCTCACAGGGTAG-3′). The relative expression of the mRNA level was normalized against the internal control GAPDH (F—5′-AAG GTCATCCCAGAGCTGAA-3′, R—5′-CTGCTTCACCACCTTCTTGA-3′) or AC TB (F—5′-GATGGTATGAAGGCTTTGGTC-3′, R—5′-TGTGCACTTTTATTG GTCTC-3′) gene (ΔCt = Ct (target gene) − Ct (internal control gene)). The relative fold change was measured by $2^{-\Delta\Delta Ct}$ formula compared with the control cells. Means and differences of the means with 95% confidence intervals were obtained using GraphPad Prism (GraphPad Software Inc.). Two-tailed Student's $t$ test was used for unpaired analysis comparing average expression between conditions. $P$-values < 0.05 were considered statistically significant.

**Gene expression analysis.** The RNA extracts were first analyzed by a Nanodrop 2000 spectrophotometer (Thermo Fisher Scientific, Waltham, MA). RNA quality was determined by the ratios of A260/A280 (close to 2) and A260/A230 (close to 2). Qualified RNAs were further tested using an Agilent 2100 Bioanalyzer (Agilent Technologies, Santa Clara, CA), and samples with RIN > 7 were selected for microarray analysis using the Affymetrix MTA 1.0 (Affymetrix). The labeling, hybridization, scanning, and data extraction of microarray were performed according to the recommended Affymetrix protocols. Briefly, the fluorescence signals of the microarray were scanned and saved as DAT image files. The AGCC software (Affymetrix GeneChip Command Console) transformed DAT files into CEL files to change image signals into digital signals, which recorded the fluorescence density of probes. Next, we used Affymetrix Expression Console software to pretreat CEL files through Robust Multichip Analysis (RMA) algorithm, including background correction, probeset signal integration, and quantile normalization. After pretreatment, the obtained chp files were analyzed by Affymetrix Transcriptome Analysis Console software to detect differentially expressed genes.

**FACS and apoptosis assay.** For analyses of T cells and gMDSC population, single-cell suspensions were prepared from blood, spleens, and lungs. Lung tissues were dissociated and digested with collagenase (Stem Cell Technologies) for 1 h at 37 °C. Red blood cells were lysed by ACK lysis buffer (Gibco). The cells were labeled with fluorescence-conjugated CD3(#100219-dilution 1/200), CD8a (#100706-dilution 1/200), CD8b(# 126608-dilution 1/200), Ly6C (#128015-dilution 1/400), and CD11b (#101208-Dilution 1/200) antibodies (Biolegend) and analyzed on a FACS canto flow cytometer (BD Biosciences). For coculture experiments, spleen-derived cells were sorted with a FACS Aria cell sorter (BD Biosciences) based on CD8 and/or Ly6C expression. Subsequently, tumor cells were cultured either alone (control) or cocultured with immune cells in 10% FBS RPMI media for 48 h at the ratios of 1:1 and 1:5. After the incubation, tumor cells were collected and Annexin V staining was performed. Immune cells were excluded by gating on CD45 (#103108-Biolegend) negative population. Annexin V (#640920-Biolegend) and cleaved-caspase-3 (#9978-Cell signaling) antibodies were used according to the manufacturer's recommendation. All tests were performed in duplicates and repeated twice.

**Immunostaining and TUNEL assay.** For immunohistochemistry, paraffin-embedded sections were de-paraffinized in xylene and rehydrated in graded alcohol. Antigen retrieval was done by incubating the sections in citrate buffer pH 6 (Invitrogen) using microwave. Staining was performed using Peroxidase Detection Kit (#1859346-Thermo scientific) with pan-keratin (# PM162AA) antibody (Biocare Medical) according to the manufacturer's protocol. For the TUNEL assay, TdT In Situ Apoptosis Detection Kit (#4811–30-K) were used according to the manufacturer's recommendation (R&D System). Briefly, rehydrated lung tissue slides were incubated with Proteinase K solution for 15 min and then washed with deionized water before immersion in quenching solution for 5 min. Following the incubation with 1X TdT labeling buffer for 5 min, tissues were kept in labeling reaction mix for 60 min at 37 °C. The reaction was stopped and samples were washed twice with PBS for 5 min. Finally, the slides were incubated with horse-radish peroxidase-conjugated streptavidin at 37 °C for 10 min and immersed in Blue label solution for 5 min. After in situ labeling, tissues were counterstained with nuclear fast red, dehydrated, and mounted. The average number of apoptotic cells were calculated by counting three independent area under bright-field microscope.

**Reporting summary.** Further information on experimental design is available in the Nature Research Reporting Summary linked to this article.

## Data availability

The data discussed in this publication have been deposited in NCBI's Gene expression Ominbus under the GEO Series accession: https://www.ncbi.nlm.nih.gov/geo/query/acc.cgi?acc = GSE81701 (GSMP190229, GSM190230). The TCGA data referenced during the study are in part based upon the data generated by the TCGA Research Network: http://cancergenome.nih.gov/ and are available in a public repository from the cBIoportal for Cancer Genomics website http://www.cbioportal.org/. All the other data supporting the findings of this study are available within the article and its Supplementary Information files and from the corresponding author upon reasonable request.

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

## Acknowledgements

We gratefully acknowledge the generous help from Flow Cytometry, Genomics Core facilities, and Labaratory of Animal Services. We thank Drs. Rafi Ahmed and Paulo C. Rodriguez for insightful discussions and comments, Dr. Iskander Asm for for helping with and training of our staff on the tail vein injections. This work was supported by start up funds to H.K. by Georgia Cancer Center. Additional research fundings to H.K. provided by American Cancer Society Institutional fund, Forbes Institute research fund, and Bridge Fund by Augusta University Research Inc.

## Author contributions

H.K., E.C. and R.P. conceived and designed the experiments, H.K., R.P., E.L. and M.O. performed the experiments, H.K., E.C. and R.P. analyzed and interpreted the data, H.K., R.P., E.C. and M.S.W. contributed to preparing the paper; R.J.B., A.H.V., A.S.A., D.M., M.G., M.T., A.C., K.A.H., E.C. and J.K.C. provided reagents.

## Additional information

**Competing interests:** The authors declare no competing interests.

