## [Peer Review File · Nature Communications]

Reviewers' Comments:

Reviewer #1:

Remarks to the Author:

In the manuscript entitled "Primary tumor-induced immunity eradicates disseminated tumor cells in syngeneic mouse model", Piranlioglu and colleagues demonstrate that while 4T1 carcinoma cells are proficient in establishing metastasis in a syngeneic mouse model, EMT6 carcinoma cells fail to do so despite of disseminating to secondary organs. They attribute this difference to T-cell mediated anti-tumor immunity, involving a subset of Ly6C +CD8+ T-cells that exhibit effector function. The authors proceed to demonstrate that complete resection of primary EMT6 tumors leads to the elimination of disseminated tumor cells (DTCs) relative to partial resection of the same tumors. While the observations reported by the authors are interesting, the lack of strong experimental evidence to support some of the conclusions, tempers the enthusiasm for this study.

Major points:

1. While the authors attribute the failure of EMT6 tumors to establish metastasis to the presence of CD8+ T-cells, the presence and potential contribution of NK cells has not been considered. Several reports in the literature have documented NK-mediated clearance of DTCs and it is likely that NK cells may also contribute to the clearance of EMT6 carcinoma cells. This maybe particularly true in the case of results obtained from Figure 7, where the depletion of CD8+ T-cells did not lead to the establishment of metastasis. Attributing only CD8+ T-cell induced anti-tumor immunity as a mechanism for clearance of EMT6 DTCs without eliminating the contribution of NK cells, weakens the conclusions from this study.

2. The authors demonstrate the presence of a CD11B- LY6C + subset as CD8+ T-cells as this subset of cells secrete cytolytic effector molecules and is significantly reduced upon treatment of tumor-bearing mice with anti-CD8. However, the antibody clone used for depletion (anti-CD8 clone 2.43) targets CD8alpha, which is also expressed by dendritic cells. Additionally, the effector molecules expressed by these cells (Interferons and granzyme B) can also be secreted by dendritic cells expressing CD8alpha. To be sure that these cells are indeed CD8+ cytotoxic T-cells and not dendritic cells, the authors should confirm that these cells express the lineage marker for T-cells, ie CD3 and lack the expression of CD11c, expressed by dendritic cells.

3. The authors allude to insufficient MDSC induction as one possible reason for the failure of EMT-6 carcinoma cells to establish distant metastasis. Although the authors demonstrate that this is likely to due to the activation of an anti-tumor immune response, insufficient MDSC induction may very well be due to the inability of EMT6 cells to secrete G-CSF, which is known to mobilize MDSCs, especially in the 4T1 tumor model. In fact, the authors have observed a difference in G-CSF production between EMT6 and 4T1 cells in their previous publication. It maybe possible that simply the expression of G-CSF could override the effects of T-cells in containing metastatic outgrowth. Therefore, it is critical to ascertain whether EMT6 cells engineered to express G-CSF establish metastasis in the presence of T-cells.

4. The authors conclude that the anti-tumor immune response observed in their model is due to the presence of T-cell memory. However, experimental demonstration of the presence of CD8+ memory T-cells is lacking. Analyzing the presence of T-cell memory markers (For eg. CD45RA, CD45RO, CCR7, CD62L) will strengthen this conclusion.

Minor points

1. In Fig. 1j, a control group where mice with resected EMT6 tumors are not subsequently challenged with anything for the same duration as the experimental group (EMT6 primed mice

challenged with EMT6 cells IV) should be included.

2. Corrections of typographical and grammatical errors will significantly improve the quality of the ms.

Reviewer #2:

Remarks to the Author:

The manuscript presents extensive data showing that the failure of the EMT6 cells to form distant organ metastases is linked to the immunoelimination of DTC. The authors present an elegant series of studies showing that the primary EMT6 tumor can generate memory T cells which will eliminate DTC. These data support a model to predict if tumor removal will protect against disease relapse from DTC, and is thus of potential high translational impact. While the study is of potential importance there are a number of important issues which need to be addressed.

1. There is no explanation as to why a similar effect is not observed in 4T1 tumors. Presumably, it may be due to their being less immunogenic or the presence of other immunosuppressive pathways which prevent the generation of memory T cells. The authors show data indicating that addition of granulocytic myeloid derived suppressor cells derived from 4T1 tumors is sufficient to allow metastasis to progress (Fig. 3). However, there are no insights presented as to what is fundamentally different between the immune system in these two tumors. At a minimum it would be important to compare T cell populations. Without some mechanistic insight into this issue it is difficult to determine what actually limits the spread of DTC cells. Without mechanistic insights into critical differences in the way these tumors engage the TME, it is difficult to know if these data can be generalized to other tumors.

2. To that end, studies using reciprocal implantation of gMDSCs from EMT6 tumors into mice bearing 4T1 tumors would indicate that there is a fundamental difference between this population in the two tumors, which contributes to metastasis.

3. There is also a lack of detail in the Figure legends. the authors provide little information and it is often very difficult to follow the data. The legends need to be expanded and additional experimental detail provided in the Methods section.

Reviewer #1 (Remarks to the Author):

In the manuscript entitled “Primary tumor-induced immunity eradicates disseminated tumor cells in syngeneic mouse model”, Piranlioglu and colleagues demonstrate that while 4T1 carcinoma cells are proficient in establishing metastasis in a syngeneic mouse model, EMT6 carcinoma cells fail to do so despite of disseminating to secondary organs. They attribute this difference to T-cell mediated anti-tumor immunity, involving a subset of Ly6C+CD8+ T-cells that exhibit effector function. The authors proceed to demonstrate that complete resection of primary EMT6 tumors leads to the elimination of disseminated tumor cells (DTCs) relative to partial resection of the same tumors. While the observations reported by the authors are interesting, the lack of strong experimental evidence to support some of the conclusions, tempers the enthusiasm for this study.

A-We thank the reviewer for favorable remarks and giving us the opportunity to address his/her constructive comments. We have added new data to the revised manuscript that increases the experimental evidence supporting our conclusions hoping that enthusiasm for our study is enhanced.

Major points

Q-1. While the authors attribute the failure of EMT6 tumors to establish metastasis to the presence of CD8+ T-cells, the presence and potential contribution of NK cells has not been considered. Several reports in the literature have documented NK-mediated clearance of DTCs and it is likely that NK cells may also contribute to the clearance of EMT6 carcinoma cells. This maybe particularly true in the case of results obtained from Figure 7, where the depletion of CD8+ T-cells did not lead to the establishment of metastasis. Attributing only CD8+ T-cell induced anti-tumor immunity as a mechanism for clearance of EMT6 DTCs without eliminating the contribution of NK cells, weakens the conclusions from this study.

A-1. We agree with the reviewer that although CD8 T cell depletion using the anti-CD8 α antibody eliminates immune clearance of DTCs resulting in development of metastasis, it does not rule out the possibility of NK cell involvement or contribution in this process. Because of the fact that CD8 α is also expressed on some NK cell and dendritic cell (DC) subsets as well, depletion of CD8 T cells using this antibody could deplete some NK and DC populations that could contribute to this process. In our revised manuscript, we not only used an anti-CD8 β antibody to more specifically deplete CD8 T cells, but also independently depleted NK cells in EMT6-primed mice using anti-asialo GM1 antibody. In doing so, we determined that DTC clearance in our EMT6-primed mouse model is CD8 T cell dependent and there was no major involvement or contribution of NK cells in this process.

Department or Team Name: Hasan Korkaya, DVM., PhD.
Assistant professor of Biochemistry and Molecular Biology

In addition, the reviewer referred to the data in Fig. 7 “where depletion of CD8+ T-cells did not lead to the establishment of metastasis”. In this experiment, we depleted CD8 T cells 60 days after the resection of the primary tumors as opposed to experiments in Fig. 3 where depletion was at the time of resection. Thus, the results demonstrate that the disseminated tumor cells had completely been eliminated by T cells early on and that the depletion of T cells at later time had no effect.

Q-2. The authors demonstrate the presence of a CD11b- LY6C+ subset of CD8+ T-cells as this subset of cells secrete cytolytic effector molecules and is significantly reduced upon treatment of tumor-bearing mice with anti-CD8. However, the antibody clone used for depletion (anti-CD8 clone 2.43) targets CD8alpha, which is also expressed by dendritic cells. Additionally, the effector molecules expressed by these cells (Interferons and granzyme B) can also be secreted by dendritic cells expressing CD8alpha. To be sure that these cells are indeed CD8+ cytotoxic T-cells and not dendritic cells, the authors should confirm that these cells express the lineage marker for T-cells, ie CD3 and lack the expression of CD11c, expressed by dendritic cells.

A-2. As shown below, we determined that majority of CD11b-Ly6C+ population express CD3 (76%) and CD8b (82%) and lacks the expression of CD11c. Moreover, 80% of this population displays Ly6C+CD8+ phenotype. More importantly, as mentioned above, depletion of CD8 T cells using anti-CD8beta antibodies, which do not deplete DCs eliminated the effect.

Q-3. The authors allude to insufficient MDSC induction as one possible reason for the failure of EMT-6 carcinoma cells to establish distant metastasis. Although the authors demonstrate that this is likely to be due to the activation of an anti-tumor immune response, insufficient MDSC induction may very well be due to the inability of EMT6 cells to secrete G-CSF, which is known to mobilize MDSCs, especially in the 4T1 tumor model. In fact, the authors have observed a difference in G-CSF production between EMT6 and 4T1 cells in their previous publication. It may be possible that simply the expression of G-CSF could override the effects of T-cells in containing metastatic outgrowth. Therefore, it is critical to ascertain whether EMT6 cells engineered to express G-CSF establish metastasis in the presence of T-cells.

Department or Team Name: Hasan Korkaya, DVM., PhD.
 Assistant professor of Biochemistry and Molecular Biology

A-3. We agree with the reviewer that tumor-secreted G-CSF plays a major role in mobilization of MDSCs in 4T1 models as we previously demonstrated (Ouzounova et al., 2017 Nature Communications). Furthermore, two independent studies by Kowanetz et al., 2010 PNAS and Waight et al., 2011 PLOS One) demonstrated that G-CSF overexpression in tumors promotes their aggressive properties/metastasis by inducing MDSC mobilization. In accordance, we have added new data showing that injection of recombinant G-CSF in EMT6 tumor-bearing mice enhances the metastatic ability of these tumors via induction of gMDSC mobilization, which suppress the CD8+Ly6C+ effector T cell population.

Q-4. The authors conclude that the anti-tumor immune response observed in their model is due to the presence of T-cell memory. However, experimental demonstration of the presence of CD8+ memory T-cells is lacking. Analyzing the presence of T-cell memory markers (For eg. CD45RA, CD45RO, CCR7, CD62L) will strengthen this conclusion.

A-4. Although reviewer's point is well taken, we convincingly demonstrated that EMT6-primed mice repeatedly rejects the tumors in subsequent challenges even after several months suggesting that it is through effector/memory T cells. The fact that we are not working with a specific tumor antigen, it is difficult to specifically identify tumor antigen specific memory T cells. Nonetheless, as shown below, we have analyzed subpopulations of CD8 T cells using CD44/CD62L surface markers as suggested by the reviewer and shown that there was a sharp increase of CD44+CD62L+ subset (mouse memory T cell phenotype) in EMT6-primed mice following the tail vein injection of EMT6-Luc cells suggesting that these may be the tumor-antigen specific memory T cells.

Minor points

Q-1. In Fig. 1j, a control group where mice with resected EMT6 tumors are not subsequently challenged with anything for the same duration as the experimental group (EMT6 primed mice challenged with EMT6 cells IV) should be included.

A-1. As requested, we have now added the Luci intensity data of the control mice into the indicated graph in Fig. 1j.

Department or Team Name: Hasan Korkaya, DVM., PhD.
Assistant professor of Biochemistry and Molecular Biology

Q-2. Corrections of typographical and grammatical errors will significantly improve the quality of the ms.

A-2. We have now corrected those typographical and grammatical errors in the revised manuscript.

Reviewer #2 (Remarks to the Author):

Q. The manuscript presents extensive data showing that the failure of the EMT6 cells to form distant organ metastases is linked to the immunoelimination of DTC. The authors present an elegant series of studies showing that the primary EMT6 tumor can generate memory T cells which will eliminate DTC. These data support a model to predict if tumor removal will protect against disease relapse from DTC, and is thus of potential high translational impact. While the study is of potential importance there are a number of important issues which need to be addressed.

A-We greatly appreciate reviewer's favorable remarks and giving us the opportunity to address his/her constructive comments in our revised manuscript.

Q-1. There is no explanation as to why a similar effect is not observed in 4T1 tumors. Presumably, it may be due to their being less immunogenic or the presence of other immunosuppressive pathways which prevent the generation of memory T cells. The authors show data indicating that addition of granulocytic myeloid derived suppressor cells derived from 4T1 tumors is sufficient to allow metastasis to progress (Fig. 3). However, there are no insights presented as to what is fundamentally different between the immune system in these two tumors. At a minimum it would be important to compare T cell populations. Without some mechanistic insight into this issue it is difficult to determine what actually limits the spread of DTC cells. Without mechanistic insights into critical differences in the way these tumors engage the TME, it is difficult to know if these data can be generalized to other tumors.

A-1. Our previously published studies (Ouzounova et al., 2017) and experiments in this manuscript provide a compelling evidence that induction and mobilization of gMDSCs suppress the anti-tumor immune responses. Our new data (Supplementary Fig. 4) also provides evidence that lungs from EMT6 tumor-bearing mice show significantly higher infiltration of CD8+Ly6C+ T cells compared to that from 4T1 tumor-bearing animals suggesting that effector T cell infiltration is prevented in metastatic tumor-bearing mice.

Q-2. To that end, studies using reciprocal implantation of gMDSCs from EMT6 tumors into mice bearing 4T1 tumors would indicate that there is a fundamental difference between this population in the two tumors, which contributes to metastasis.

A-3. Even though we already published gMDSCs from 4T1 tumor-bearing mice displayed significantly higher suppressive ability compared to the EMT6 tumor-bearing mice (Ouzounova et al., 2017), we performed these studies early on including the reciprocal implantation of MDSCs in EMT6 or 4T1 tumor-bearing mice. However, as we expected MDSCs from EMT6-tumor-bearing mice had no effect on 4T1 tumor growth and metastasis. As shown below, co-injection of gMDSCs or mMDSCs from EMT6 tumor-bearing mice with 4T1-Luc cells in 4T1- primary tumor-bearing mice had no noticeable effect, tail vein injected 4T1-luc cells generated pulmonary metastasis within two weeks, irrespective of MDSCs from EMT6 tumor-bearing mice.

Department or Team Name: Hasan Korkaya, DVM., PhD.
Assistant professor of Biochemistry and Molecular Biology

Mailing Address:
1410 Laney Walker Blvd. CN2136
Augusta, Georgia 30912

Office Address:
1410 Laney Walker Blvd. CN2136
Augusta, Georgia 30912

T 706-721-2429
F 706-721-0469

cancer.augusta.edu

We did not include these data because we believe that they do not add any value to the manuscript but will be happy to include it as supplementary data if the reviewer feels is necessary to do so.

Q-3. There is also a lack of detail in the Figure legends. The authors provide little information and it is often very difficult to follow the data. The legends need to be expanded and additional experimental detail provided in the Methods section.

A-3. We now addressed this issue in the revised manuscript as suggested by the reviewer.

We appreciate the thoughtful comments on our manuscript by the reviewers. We believe that these modifications have significantly improved the quality of manuscript and hope that is now suitable for publication in *Nature Communications*.

Sincerely

Hasan Korkaya, DVM., PhD.

Department or Team Name: Hasan Korkaya, DVM., PhD.
Assistant professor of Biochemistry and Molecular Biology

Mailing Address:
1410 Laney Walker Blvd. CN2136
Augusta, Georgia 30912

Office Address:
1410 Laney Walker Blvd. CN2136
Augusta, Georgia 30912

T 706-721-2429
F 706-721-0469

cancer.augusta.edu

Reviewers' Comments:

Reviewer #1:

Remarks to the Author:

The authors have thoughtfully addressed all the concerns raised in the previous version of the manuscript. The revised manuscript demonstrates compelling findings and seems suitable for publication in its current form

Reviewer #2:

Remarks to the Author:

This revised manuscript support a role for a subset of CD8 T cells in mediating elimination of DTC using a relevant mouse model. The authors have been responsive to the previous reviewers' comments and have performed several additional experiments to support their model. In particular studies addressing the role of NK cells have been addressed. While it is conceivable that other cell types participate in the elimination of DTCs in the EMT6 tumors, the data support a major role for CD8 Tcells.

Overall the authors have been very responsive to the comments of the reviewers, and the manuscript is markedly improved.

Reviewer #1 (Remarks to the Author):

The authors have thoughtfully addressed all the concerns raised in the previous version of the manuscript. The revised manuscript demonstrates compelling findings and seems suitable for publication in its current form

Reviewer #2 (Remarks to the Author):

This revised manuscript support a role for a subset of CD8 T cells in mediating elimination of DTC using a relevant mouse model. The authors have been responsive to the previous reviewers' comments and have performed several additional experiments to support their model. In particular studies addressing the role of NK cells have been addressed. While it is conceivable that other cell types participate in the elimination of DTCs in the EMT6 tumors, the data support a major role for CD8 Tcells.

Overall the authors have been very responsive to the comments of the reviewers, and the manuscript is markedly improved.

We are extremely grateful to the reviewers for their endorsement of our manuscript for publication after addressing their comments/critiques in the first round of review process.